# Machine Learning Models Combined with Virtual Screening and Molecular Docking to Predict Human Topoisomerase I Inhibitors

**DOI:** 10.3390/molecules24112107

**Published:** 2019-06-04

**Authors:** Bingke Li, Xiaokang Kang, Dan Zhao, Yurong Zou, Xudong Huang, Jiexue Wang, Chenghua Zhang

**Affiliations:** 1Institute of Functional Molecules, College of Chemistry and Life Science, Chengdu Normal University, Chengdu 611130, China; libingke86@126.com (B.L.); KXK2799956460@163.com (X.K.); audreyzhaodan@163.com (D.Z.); zyrlia1018@163.com (Y.Z.); hxd1997116@163.com (X.H.); 2School of Biological and Chemical Engineering, Nanyang Institute of Technology, Nanyang 473004, China; 3School of Basic Medical Sciences, North Sichuan Medical College, Nanchong 637000, China

**Keywords:** machine learning, virtual screening, human topoisomerase I, inhibitors, molecular descriptors, molecular docking

## Abstract

In this work, random forest (RF), support vector machine, k-nearest neighbor and C4.5 decision tree, were used to establish classification models for predicting whether an unknown molecule is an inhibitor of human topoisomerase I (Top1) protein. All these models have achieved satisfactory results, with total prediction accuracies from 89.70% to 97.12%. Through comparative analysis, it can be found that the RF model has the best forecasting effect. The parameters were further optimized to generate the best-performing RF model. At the same time, features selection was implemented to choose properties most relevant to the inhibition of Top1 from 189 molecular descriptors through a special RF procedure. Subsequently, a ligand-based virtual screening was performed from the Maybridge database by the optimal RF model and 596 hits were picked out. Then, 67 molecules with relative probability scores over 0.7 were selected based on the screening results. Next, the 67 molecules above were docked to Top1 using AutoDock Vina. Finally, six top-ranked molecules with binding energies less than −10.0 kcal/mol were screened out and a common backbone, which is entirely different from that of existing Top1 inhibitors reported in the literature, was found.

## 1. Introduction

Supercoiling, knotting and catenation—three main types of topology—keep DNA firmly compacted into chromatin [1]. Nevertheless, excessive supercoiling can seriously hinder replication and transcription that alters the DNA structure at inopportune times [2]. Therefore, transient unwinding and loosening of the parent supercoiled DNA are very crucial in order to maintain the integrity of the genetic material when a cell divides [3]. Topoisomerases (Tops) are essential and ubiquitous DNA processing enzymes that can deal with various topological issues through regulation of the super torsional strains generated during a series of vital cellular metabolic processes, including not only replication and transcription, but also repair, recombination and segregation of DNA, in conjunction with chromatin assembly, and so on [4,5,6].

Tops are classified as two general subfamilies, type I and type II, based upon their mechanisms of action [7,8,9]. Furthermore, each category can be broken down into subtypes A or B. The type IB Tops, represented by human topoisomerase I (Top1), is a single-gene-encoded monomeric protein 765 amino acids in length, consisting of four domains: N-terminal domain, linker domain, core domain, and C-terminal domain. The N-terminal domain is charged and extremely disordered, but it contains nuclear localization signals. The linker domain connects the core domain and C-terminal domain. The first two are dispensable for catalytic activity. The large core domain appears to be highly conserved, possessing four (Arg488, Lys532, Arg590, and His632) out of five of basic catalytic residues which are responsible for DNA binding. The last catalytic residue (Tyr723) is in the C-terminal domain.

Top1 operates using a “hindered rotation mechanism”, which involves three major steps [10,11,12]: (1) The hydroxyl group in active site Tyr723 as a nucleophile attacks the phosphodiester bond of one DNA single strand, triggering a reversible transesterification reaction. In such a manner, a temporary intermediate Top1-DNA covalent cleavage complex (Top1cc) is generated. (2) This Top1cc regularly keeps going for enough time to allow the cracked single strand to rotate around the other, as a way to relieve superhelical tension in duplex DNA. (3) A similar transesterification reaction occurs between the hydroxyl group of the broken DNA strand and the phosphotyrosine bond in Top1cc to restore the DNA phosphodiester backbone, ensure DNA integrity, and liberate Top1 for another round of cleavage/religation reactions. Top1 is over-expressed in several human neoplasms except normal tissues, so it can be presumed that the rapidly proliferating cancer cells are more closely related to Top1 than the healthy cells [13,14]. Thus, Top1 has been recognized as a remarkably promising target in designing and developing chemotherapeutic drugs for anticancer treatments.

Top1 suppressors can be categorized into two groups: Poisons and catalytic inhibitors [15,16]. Poisons permit Top1-mediated DNA cleavage, but selectively trap, stabilize and freeze Top1cc for preventing DNA resealing, which creates a locked ternary complex of drug, protein and cleaved DNA to transform a functional enzyme into a lethal component. Unrepaired Top1cc causes permanent DNA double strand breaks when it collides with the DNA replication fork, ultimately leading to cell cycle arrest and apoptotic cell death [17]. In contrast, catalytic inhibitors directly bind to enzymes, but do not participate in the stabilization of Top1cc, thereby hindering other processes of Top1 catalytic cycle.

A large number of small molecule inhibitors aimed at Top1 have proven biologically active and clinically effective. Among them, the most potent two semisynthetic camptothecin (CPT) derivatives, irinotecan and topotecan, are approved by U.S. Food and Drug Administration as drugs for cancer chemotherapy [18,19]. Irinotecan has been widely prescribed as first-line therapy for metastatic colorectal cancer, whereas topotecan is currently used as second-line therapy for ovarian, cervical and small cell lung cancers. Despite the clinical success of conventional CPT derivatives, significant hurdles, such as dose-limiting toxicities and drug resistance, still exist in the use of these drugs [12,20].

To overcome the limitations and drawbacks associated with CPT, various non-CPT inhibitors with a structurally unique scaffold, for instance, indolocarbazoles, indenoisoquinolines and benzophenanthridines as poisons [21], coupled with diarylisoquinolines and porphyrins as catalytic inhibitors [15,16], have been developed, some of which are even being put into clinical practice, raising the possibility that another novel class of Top1 inhibitors (Top1is) can be marketed in the future. At the same time, structure-based and computer-aided drug design have made enormous efforts with the publication of Top1 crystal structures [22] to the discovery, design and development of anti-cancer candidates targeting Top1. Feng et al. [23] constructed pharmacophore-guided 3D-QSAR models based on evodiamine analogs, which are one category of Top1is, obtained ten well-predicted compounds through virtual screening and investigated the action mode of protein-ligand by molecular docking and molecular dynamics. Thai et al. [24] performed a support vector machine (SVM) classification model on a suite of 73 Benzo[*c*]phenanthridine derivatives with Top1 inhibitory activity, achieving a total prediction accuracy (Q) of 87% and a Matthews correlation coefficient (MCC) of 0.71 by a testing set of 15 compounds.

In the present study, four machine learning (ML) approaches were used to build up classification models and virtual screening technology applied for searching potential Top1is with new structures were combined. These methods are random forest (RF) [25], SVM [26], k-nearest neighbor (k-NN) [27], and C4.5 decision tree (DT) [28]. In a subsequent step, modes of action about Top1 and selected inhibitors were investigated via molecular docking. In the final step, six compounds with unusual scaffold and the highest scores were screened out. Our research findings not only prove that the strategy of ML classification models combined with virtual screening and molecular docking is reliable, but also provide a remarkable theoretical basis for the further chemical synthesis, structural characterization and biological testing of the six new possible Top1is.

## 2. Results and Discussion

### 2.1. Structural Diversity Analysis

The structural diversity of data set A can be estimated by *D*(*A*), which represents an average in the degree of dissimilarity among all pairs of compounds:(1)$$D(A)=\frac{\sum_{i=1j=1}^{N}\sum_{i\neq j}^{N}diss(i,j)}{N(A)[N(A)-1]}$$
where *N*(*A*) is the number of compounds in the data set A, and *diss*(*i*, *j*) is a measure of the dissimilarity between compounds *i* and *j*. The higher the value of *D*(*A*) is, the better the structural diversity of the data set A is, and the larger the applicability domain of a model will be.

In this study, the computed *D*(*A*) values are 0.4716, 0.4824 and 0.4494 for the whole data set, the training set and the testing set, respectively, which are much higher than that of the external validation set in recent literature [29], and also superior to that of the data set in our previous work [30], as shown in Table 1. These results reflect considerable structural diversity for our data sets. In the meantime, the *D*(*A*) value of molecules picked out by virtual screening in this work was also computed. A relatively low value of 0.1281 may be mainly attributed to the similarity of the histone deacetylase (HDAC)_Library in the Maybridge database, whose *D*(*A*) value is only 0.1550.

### 2.2. Comparison on Prediction Accuracies of Four Machine Learning Methods

Four methods, RF, SVM, k-NN and C4.5 DT were used to develop classification prediction models for the same training set in this study, and the prediction performances of the models were measured with the same testing set. The data are summarized in Table 2.

*M_try_* (the number of randomly preselected variables in each tree), σ and *k*, which are the parameters of RF, SVM and k-NN methods, respectively, were confirmed by the internal selection programs. TP (true positive) denotes the number of actives predicted correctly, TN (true negative) denotes the number of inactives predicted correctly, FP (false positive) denotes the number of inactives mispredicted as actives, and FN (false negative) denotes the number of actives mispredicted as inactives. At the same time, there are several accuracy measures to evaluate the prediction performance, including sensitivity (SE), specificity (SP), Q and MCC [31]. These measures have the following relationships with the variables aforementioned:(2)$$SE=\frac{TP}{TP+FN}$$
(3)$$SP=\frac{TN}{TN+FP}$$
(4)$$Q=\frac{TP+TN}{TP+TN+FP+FN}$$
(5)$$MCC=\frac{TP\times TN-FN\times FP}{\sqrt{\left(TP+FN\right)\left(TP+FP\right)\left(TN+FN\right)\left(FN+FP\right)}}$$

It can be seen from Table 2 that the Q value of the RF model is the highest among the four kinds of models, and the MCC value of the RF model is also the largest. In addition, they far outweigh the literature values [24]. These phenomena give expression to greater advantages for the RF method compared with the other three and the literature.

### 2.3. Optimization of RF Model Parameters

To achieve higher accuracy and better performance in the four methods, the optimum parameters of the RF method were obtained by tuning and combining two basic RF parameters, *M_try_* and *N_tree_* (the number of trees generated). Different values for these two parameters were tried continually until the prediction error rate (PER) of out-of-bag (OOB) for the testing set achieved a relatively low value.

The forecast effects in different RF models constructed by different values of parameters *M_try_* (see in Figure 1A) and *N_tree_* (see in Figure 1B and Appendix A) were studied synergistically. From the histograms, it can be expressed more intuitively. There are various forecasting models established by the RF method with diverse parameters. For Figure 1A, when the value of *M_try_* was set to 15, the PER of OOB in the testing set reaches the lowest of 2.88%, so 15 was chosen as the optimal solution of this parameter. Since the parameter *M_try_* had been determined, the *N_tree_* value was constantly changed in order to get a model with best performance, and simultaneously the *M_try_* value remained unchanged at 15.

Figure 1B and Appendix A illustrate that the testing set has the lowest OOB PER (2.88%) if the *N_tree_* value is located in one of the intervals 180–230, 480–560 or 1580–1610. However, when the *N_tree_* value is taken from 181 to 184, the corresponding training set has the PER of 8.24%, which is lower than the others. Besides, the greater the *N_tree_* value, the slower the computation speed, so 181 serves as the most suitable parameter. In conclusion, when the *M_try_* and *N_tree_* values of the RF method are fixed at 15 and 181, respectively, the corresponding model has the best prediction effect.

### 2.4. Evaluation of RF Optimal Model

For the established RF optimal model, Figure 2 shows the visualized distributions of 971 molecules in the training set and 486 molecules in the testing set. From the graph, the classification boundary lines of the model can separate Top1is from Top1 non-inhibitors (non-Top1is) well. In the testing set, three actual non-Top1is above the classification boundary line were mispredicted as Top1is, while eleven actual Top1is below the classification boundary line were mispredicted as non-Top1is, indicating that the model is not 100% accurate. It is difficult for the model to make correct predictions on these fourteen molecules. These molecules with erroneous predictions are listed in Appendix A for reference.

Furthermore, the discriminant effect of a binary classification model can also be analyzed and evaluated by plotting the receiver operating characteristic (ROC) curve [32]. ROC curve combines SE and SP to identify how the model performs. As prediction probability threshold changes, a panel of SE and “1-SP” will be worked out. If SE is used as dependent variable, and simultaneously “1-SP” is viewed as independent variable, the ROC curve could be graphed by connecting each point in turn. When the prediction probability threshold is continuously changed, the points on the curve stand for a trade-off between SE and SP. There is also a particularly important index to assess the prediction ability of a classification model: The area under the ROC curve (AUC), whose value is between 0.5 and 1. To be more precise, the larger the AUC value, the better the model performance. The ROC curves of the optimal RF model for the training set and the testing set are shown in Figure 3. The computed AUC values of the training set and the testing set are 0.968 and 0.989, respectively, which proves the excellent precision of the RF model. In order to further verify the prediction performance of the above model, an external validation set not involved in the internal data sets was assayed under the same training condition. As a result, the optimal RF model perfectly forecasted 55 inhibitors with known Top1 activities for one hundred percent Q. The visualized distributions of the 55 molecules in the external validation set are depicted in Appendix A.

### 2.5. Features Selection

By means of the special procedure of the RF method, the model associated with the optimal parameters was also processed in the aspect of features selection. From the 189 descriptors (see Section 3.2), ten descriptors which are most relevant to the properties of Top1is were screened out, as listed in Figure 4. Each of these 189 descriptors has its corresponding contribution rate. For the sake of contrastive analysis, 63 descriptors are arranged in Appendix A based on its relative importance.

In Figure 4, the significance of the 10 descriptors decreases successively. Specifically, Rugty, Tcent and S(27), namely, molecular folding, central index, and :C:: sum of atom-type electrical topological states, rank as the top three. Analogously, 63 descriptors and 37 descriptors were extracted separately from the SVM model and the C4.5 DT model, as ranked in Appendix A. For the SVM model, the top three descriptors are ^5^χ_CH_ (Simple molecular connectivity Chi indices for cycles of 5 atom), Rugty and Tcent, the last two of which are identical with the first two descriptors in the RF model. For the C4.5 DT model, S(27) ranks first, which is the same as the third descriptor in the RF model. So it can be deduced that the three characteristics of Rugty, Tcents and S(27), have especially crucial reference values in predicting potential Top1is. Among the models RF, C4.5 DT and SVM, 15 descriptors were selected collectively by all of the three, while 41, 21 and 23 descriptors were chosen by two of the three, as emerged in Figure 5.

### 2.6. Virtual Screening

The aforementioned RF optimal model was used for virtually screening 4107 compounds from the HDAC_Library in the Maybridge database, and finally 596 hits were picked up. It can be recognized that many compounds have the same basic skeleton structure, such as compounds MBX026907 and MBX026908, together with compounds MBX114890 and MBX114891, as shown in Figure 6.

The results show that the RF optimal model has filtered a diverse set of helpful structures from the database. However, due to the excessive number of screened molecules, further processing was needed, so 67 molecules with RF scores greater than 0.7 were picked out, as detailed in Appendix A, together with the visualized distributions graph drawn in Appendix A.

### 2.7. Docking Calculation

A total of 67 molecules were docked in batches with AutoDock Vina [33] for further identification of small molecules that are more likely to become drugs. The docking experiments were carried out in the grid box of receptor protein to produce the optimal conformation of docking and the corresponding ligand-receptor binding energy for each ligand. The docking binding energies of the 67 molecules are specified in Appendix A.

By ranking the 67 molecules on the basis of their docking binding energies, we screened out six top-ranked molecules with binding energies less than −10.0 kcal/mol as potential Top1is, according to the principle that the lower the binding energy, the more stable and intimate the ligand-receptor interaction. The comparison results of molecular structures and binding energies for the first six ligands and the original ligand are displayed in Table 3.

It can be found that the binding energy of the original ligand with receptor protein is −9.0 kcal/mol, while the top six ligands with best binding affinity are shown by MBX534706, MBX162127, MBX209152, MBX161748, MBX161745 and MBX190732 with docking scores from −11.4 to −10.1, which are smaller than that of the original ligand, suggesting higher potential inhibitory activities for them towards Top1. Some of the six molecules, MBX161748, MBX161745 and MBX190732, share the same basic skeleton, which is demonstrated in Figure 7. The structure of the common backbone is entirely different from that of existing Top1is reported in the literature, meaning that the scaffold may be a new original nucleus helpful for the suppressing of Top1 and cancer therapy.

The conformations and interactions between receptor protein and the preceding 67 ligands were analyzed further by AutoDockTools-1.5.6, among which some typical samples are clearly exemplified in Figure 8. Ligand MBX534706 in Figure 8A, whose carbonyl oxygen forms hydrogen bond with amino acid residue ASN352 in Top1, can bond stably with the surrounding base DT10 in DNA. Ligand MBX162127 in Figure 8B, can not only interact with bases TGP11 and DA113 in DNA, but also has relevance to amino acid residues LYS425 and TYR426 in Top1. Ligand MBX161745 in Figure 8C, has a carbonyl oxygen forming hydrogen bond with base DA113 in DNA. The primitive ligand in Figure 8D, are associated with groups such as DA113, DT10 and TGP11 in DNA, proving that our ligands winnowed from docking dovetail beautifully with the original ligand in binding mode.

It is obvious from the above that the results from molecular docking after ligand-based virtual screening are credible and feasible. The six selected ligands are most likely Top1is. Nevertheless, they are short of experimental validation. Further research and verification are needed, such as chemical synthesis, structural characterization, biological tests in vitro, etc.

## 3. Materials and Methods

### 3.1. Data Sources

A total of 1457 compounds were collected, ranging in molecular weight from 117 to 2091 Da (more than 70% of compounds with molecular weights in the range of 300 to 600 Da, as in Figure 9).

Among the compounds, 729 Top1is (labeled by “1”) with corresponding biological activity data were retrieved from the Thomson Reuters Integrity database (https://integrity.thomson-pharma.com), and for a virtually balanced 1:1 class distribution, 728 non-Top1is (labeled by “−1”) were extracted from the MDL Drug Data Report (http://www.mdli.com, MDDR) database via k-means clustering [34].

To facilitate modeling, the entire data set was randomly separated into two thirds as a training set (971 molecules, 481 Top1is and 490 non-Top1is), in combination with one third as a testing set (486 molecules, 248 Top1is and 238 non-Top1is) according to their distributions in the chemical space [35]. The training set was created with the purpose of developing a statistical model and optimizing the parameters of the ML algorithm. The testing set was used to evaluate the prediction accuracy of the model.

To further verify the performance of classification models, an external validation set including 55 Top1is was abstracted from the recent literature [29]. The molecular weights of the compounds in the external validation set and the molecules picked out by virtual screening with RF scores greater than 0.7 are all kept at reasonable levels, ranging from 200 to 500 Da, which can be seen in Appendix A. No repetition was found between molecules in the four datasets by means of similarity search.

### 3.2. Molecular Descriptors

A total of 189 molecular descriptors, which icnluded 18 simple molecular properties, 22 quantum chemical properties, 25 geometrical properties, 27 molecular shape and connectivity properties, and 97 electro-topological state properties, were applied to calculate structural and physicochemical characteristics of compounds in data sets. The calculation of these descriptors in the current study relied on the 3D structure of each agent. The descriptors computing program was written in the Fortran 77 language by our laboratory, which is available for running under Linux environment. Detailed descriptions of the 189 molecular descriptors are shown in Appendix A. Corina Symphony software (version 1.0, https://www.mn-am.com/products/corinasymphony) was applied to calculate 3D coordinates of atoms and eliminate counter ions and salts from molecular structures, by which molecules were neutralized, mesomerized and aromatized with the default parameter value.

### 3.3. Machine Learning Methods

In this research, k-NN, C4.5 DT, RF and SVM were utilized to distinguish the Top1is from non-Top1is. More information about the four methods can be easily acquired in the literature [25,26,27,28]. Thus, there are only brief descriptions for them here.

k-NN [27] is a method for classifying test cases based on the majority voting principle in the feature space, or rather, if a sample has k-nearest neighbors, most of which belong to a certain category, it can be inferred that this sample also belongs to this category.

C4.5 DTs [28] consist of leaf nodes, non-leaf nodes and intricate branches. Each non-leaf node represents a test to be conducted on a single feature value and each branch contributes to the attribute in scope of a particular output, while each leaf node is linked with a decision result. The decision process begins with the root node, then judges which output branch to choose, until it reaches a leaf node.

RF [25] is an ensemble of numerous unpruned DTs that have no dependencies on each other. Separate bootstrap samples of the training data and a set of randomly selected variables (*M_try_*) were used to determine the best possible split of each node in the tree induction. Each tree gets as much terminal growth as possible and gives its own prediction for every input data. Consequently, all the trees (*N_tree_*) generate a forest, and make a final prediction by consensus voting. An unbiased OOB estimate, which is regarded as an excellent measure equivalent to cross-validation, can internally evaluate the generalization error of RF.

SVM [26], whose main idea springs from the structural risk minimization principle, is a classifier that can map data sets from the original input space into a high-dimensional feature space, where data which are not linearly separable in low-dimensional space can be easily divided by building a decision boundary—a hyperplane.

The k-NN, C4.5 DT, RF and SVM models were generated by means of self-compiled programming written in the Fortran77 language.

### 3.4. Virtual Screening

The virtual screening technology was adopted in this work based on structures of small molecular ligands and the best ML model. The screened target database is Maybridge database (https://www.maybridge.com/), which is a small molecular database for free. The product portfolio of this database provides a comprehensive scope of chemical products and services for drug discovery and biotechnology departments.

### 3.5. Molecular Docking

The docking simulations were performed using the AutoDock Vina program downloaded from the Molecular Graphics Laboratory of the Scripps Research Institute [33]. AutoDock Vina is an open-source program used for protein-ligand docking and structure-based virtual screening due to its relative higher speed than many other docking tools, which calculates the grid maps and clusters the results automatically.

### 3.6. Targets Selection and Preparation

3D X-ray crystal structure of Top1 in complex with CPT (pdb id: 1T8I, resolution = 3 Å) was gained from the protein data bank (https://www.rcsb.org/). First, we downloaded file with a suffix “.pdb”, which can be recognized by AutoDockTools. Then, a grid box of size 16 × 16 × 16 Å with coordinates X = 22.599, Y = −2.481 and Z = 28.0 was selected to ensure the original ligand CPT was completely encapsulated. Next, hydrogen atoms were added, water residues were removed, and the original ligand in the crystal was separated from the receptor protein. Finally, through optimization the final energy minimized receptor was saved as a “.pdbqt” file for further use.

### 3.7. Ligands Preparation

The 3D structure of each small molecular ligand was calculated by Chem3D Pro 14.0. The ligands were processed with minimize energy module of MM2 procedure in Chem3D, and subsequently saved as “mol2” files. Raccoon 1.0 software, which can batch molecules, hydrogenate charges and convert formats, was used to transform ligands into “.pdbqt” files that could be identified by AutoDock Vina.

## 4. Conclusion

Top1 is currently a hot topic in the research of cancer treatment. Screening, designing and synthesizing effective Top1is are of great and far-reaching significance for anti-cancer research.

To identify the active or inactive property of a compound targeting Top1, four ML classification models (RF, SVM, k-NN and C4.5 DT) were developed in this study. Those models were compared based on several accuracy measures and the RF model outperformed others by internal OOB estimate. Further statistical evaluation, features selection and external validation were performed in succession for the optimal RF model.

Subsequently, ligand-based virtual screening was integrated with relative probability scores in the RF model to choose hits from the Maybridge database. Taking into consideration the actual condition of protein-ligand binding, we carried out molecular docking and interaction analysis for each selected ligand towards Top1. According to the ranking of computed binding energies, the first six potential inhibitors were screened out, and three of them share a new common skeleton, which has not been reported yet.

The above experimental facts demonstrate that ML classification models, combined with virtual screening and molecular docking, can greatly improve the efficiency on the aspect of discovering potential inhibitors with fascinating activities for Top1, which can be generalized to other human diseases for a certain target.

## Figures and Tables

**Figure 1 molecules-24-02107-f001:**
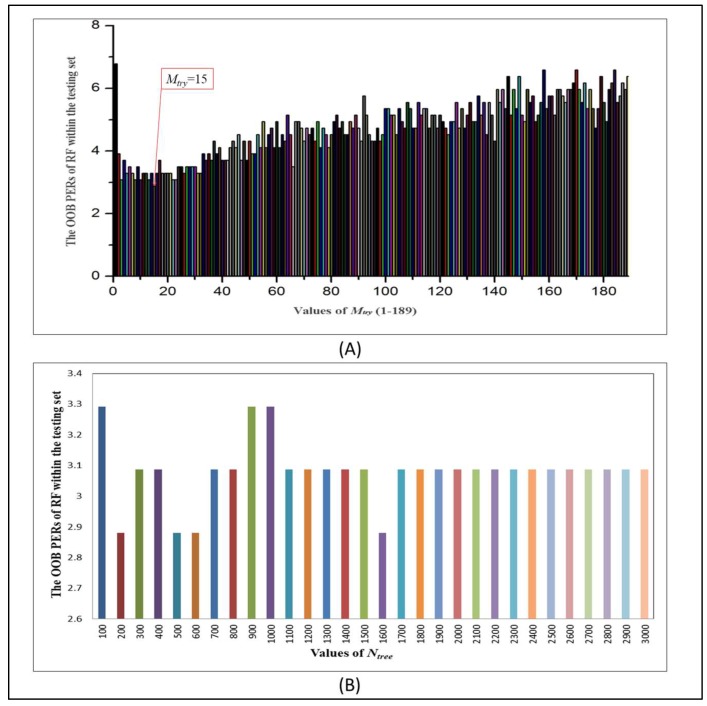
The effects of the different parameters on the out-of-bag (OOB) prediction error rates (PERs) of random forest (RF) within the testing set. (**A**) *M_try_* (1 ≤ *M_try_* ≤ 189); (**B**) *N_tree_* (100 ≤ *N_tree_* ≤ 3000).

**Figure 2 molecules-24-02107-f002:**
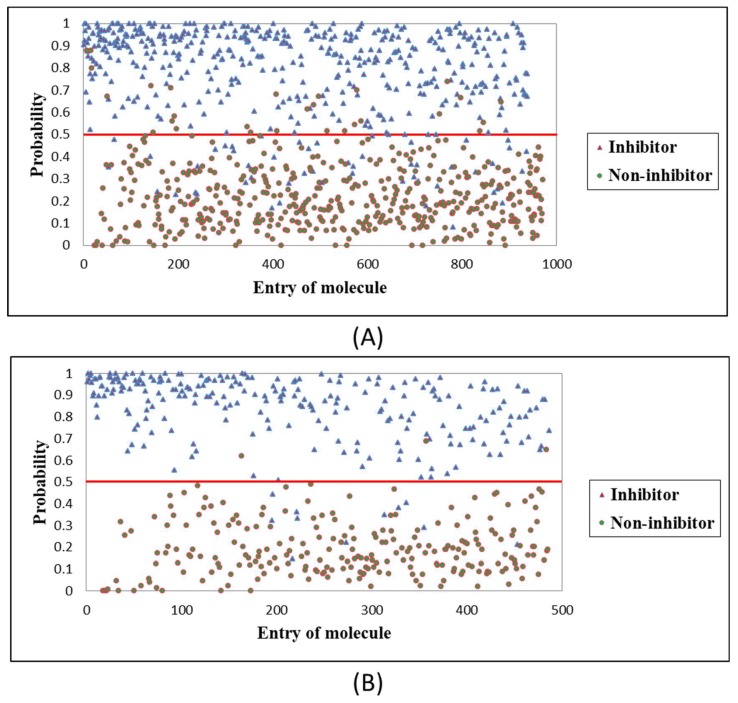
The visualized distributions of compounds, (**A**) in the training set (971 compounds), and (**B**) in the testing set (486 compounds).

**Figure 3 molecules-24-02107-f003:**
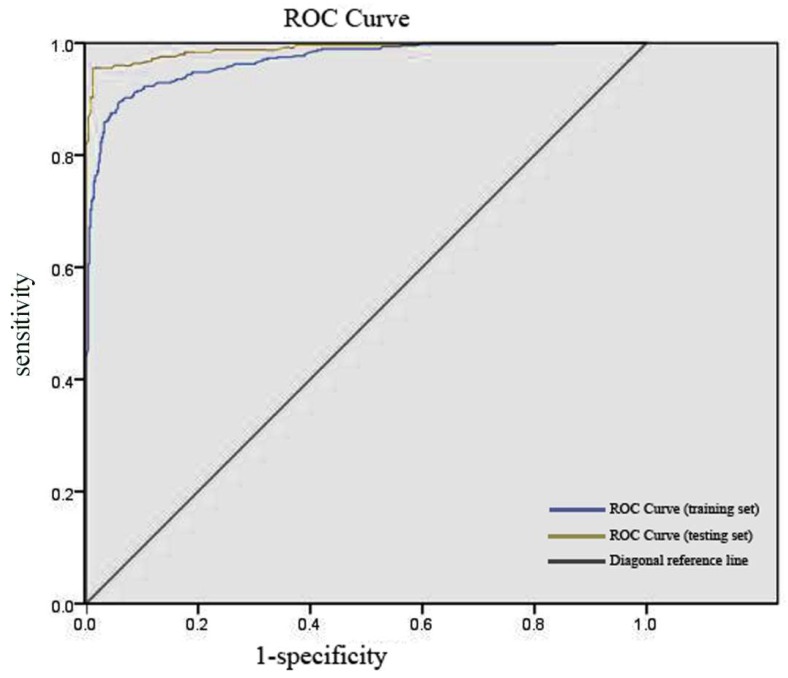
The receiver operator characteristic (ROC) curves for the optimal random forest (RF) model.

**Figure 4 molecules-24-02107-f004:**
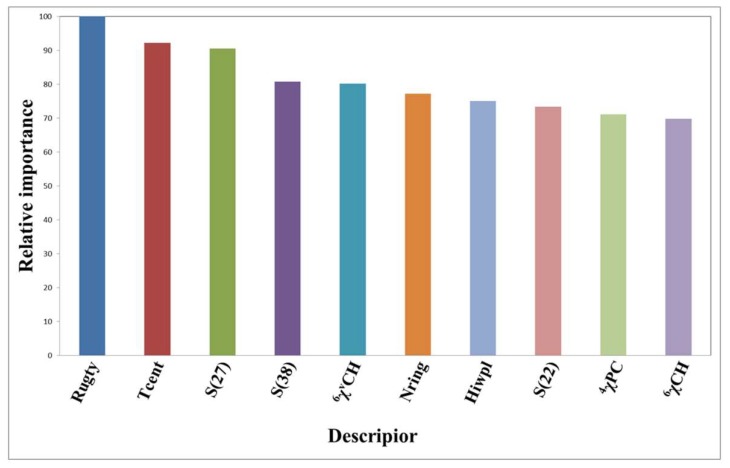
The relative importance of the 10 highest ranking descriptors in the optimal random forest (RF) model.

**Figure 5 molecules-24-02107-f005:**
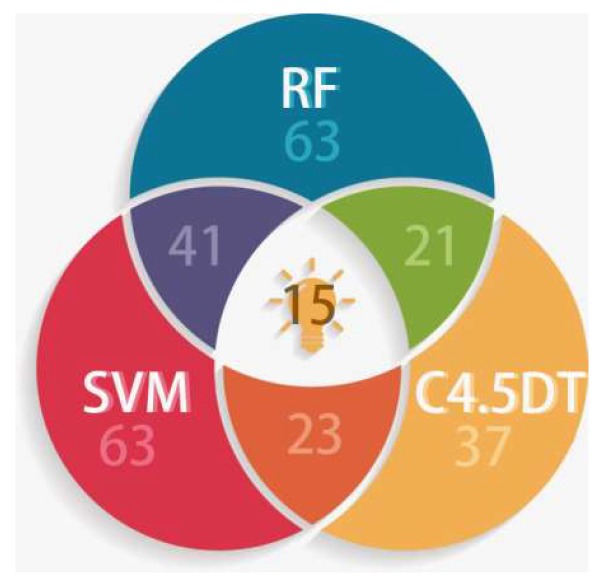
The number of descriptors selected by the models random forest (RF), C4.5 Decision Tree (DT) and support vector machine (SVM).

**Figure 6 molecules-24-02107-f006:**
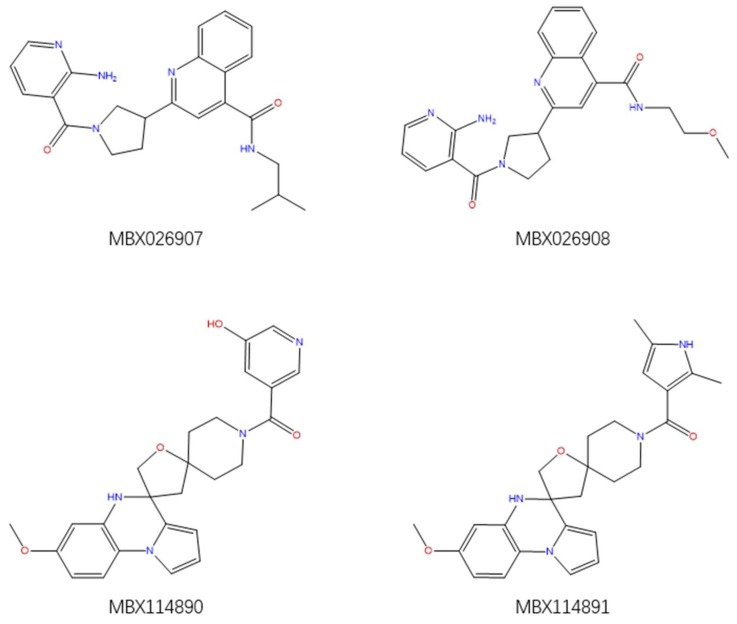
Partial selected molecules.

**Figure 7 molecules-24-02107-f007:**
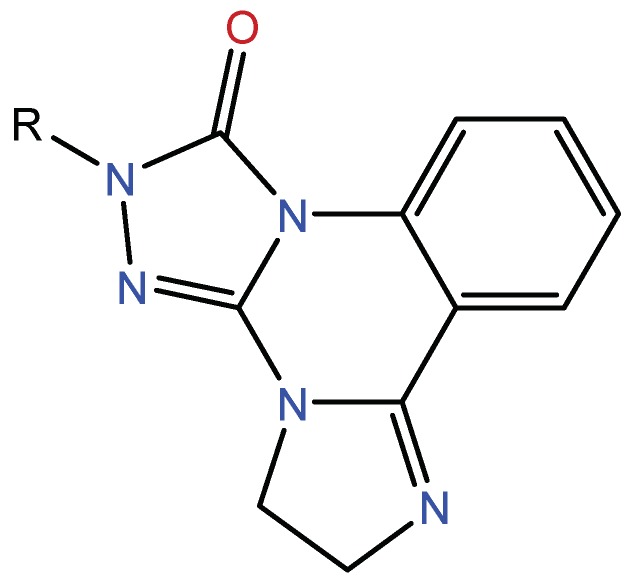
The common skeleton.

**Figure 8 molecules-24-02107-f008:**
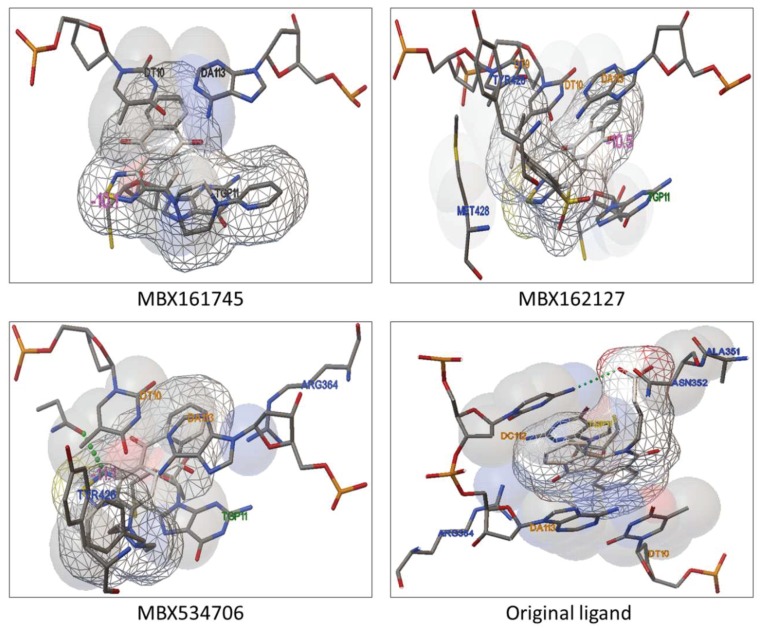
The conformations and the interactions between receptor protein and some typical ligands.

**Figure 9 molecules-24-02107-f009:**
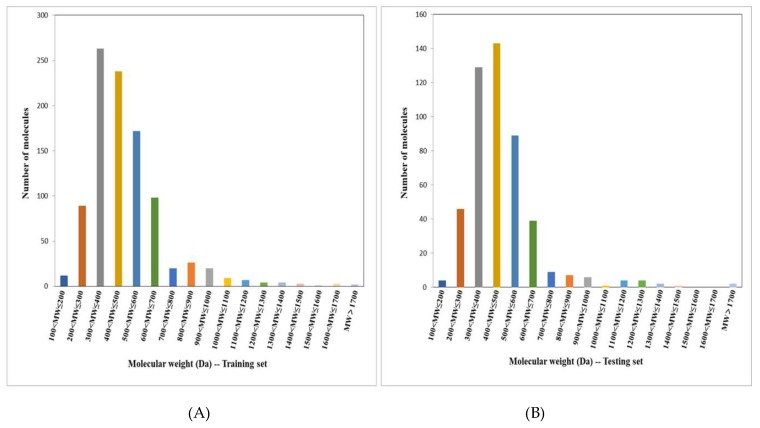
The molecular weight distributions of compounds (**A**) in the training set (971 compounds), and (**B**) in the testing set (486 compounds).

**Table 1 molecules-24-02107-t001:** The *D*(*A*) values of compounds in several data sets.

Data Sets	Number of Compounds	*D*(*A*) Values
The whole data set in this work	1457	0.4716
The training set in this work	971	0.4824
The testing set in this work	486	0.4494
The external validation set in recent literature [29]	55	0.0936
Molecules picked out by virtual screening with RF scores greater than 0.7 in this work	67	0.1281
The HDAC_Library in the Maybridge database	4107	0.1550
The whole data set in our previous work [30]	283	0.3537

**Table 2 molecules-24-02107-t002:** The comparison on the prediction accuracies of Top1 inhibitors (Top1is) and Top1 non-inhibitors (non-Top1is) from models random forest (RF), support vector machine (SVM), k-nearest neighbor (k-NN) and C4.5 decision tree (DT) by using the same testing set.

Methods	Parameters	Top1is			non-Top1is			Q (%)	MCC
		TP	FN	SE (%)	TN	FP	SP (%)		
RF	*M_try_* = 15	237	11	95.56	235	3	98.73	97.12	0.9429
SVM	*σ* = 0.2	228	20	91.94	226	12	94.96	93.41	0.8688
k-NN	*k* = 6pr	223	25	89.92	221	17	92.86	91.36	0.8277
C4.5 DT	/	223	26	89.56	212	25	89.84	89.70	0.7939

**Table 3 molecules-24-02107-t003:** Six top-ranked molecules with binding energies less than −10.0 kcal/mol.

Name	Binding Energy kcal/mol	Structural Formula
MBX534706	−11.4	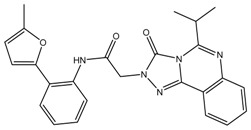
MBX162127	−10.5	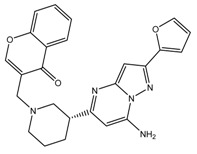
MBX209152	−10.3	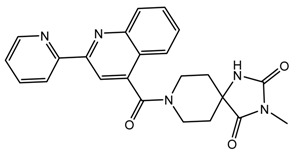
MBX161748	−10.2	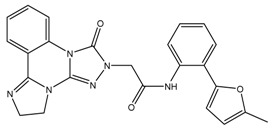
MBX161745	−10.1	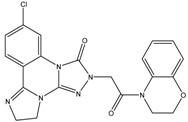
MBX190732	−10.1	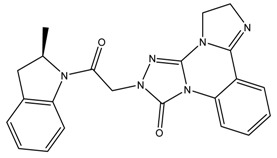
The original ligand	−9.0	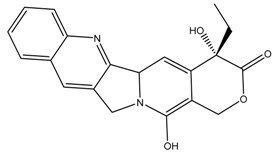

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
