# Peer review of "Machine Learning Models Combined with Virtual Screening and Molecular Docking to Predict Human Topoisomerase I Inhibitors"

_molecules, 2019, doi:10.3390/molecules24112107_

Round 1

Reviewer 1 Report

Authors present their work on computational approaches to identify Topoisomerase I Inhibitors. It is interesting work, however too limited, therefore I can not recommend it for publication.

At the end authors use Autodoc Vina for selecting final compounds, which is very standard approach.

Selected compounds are then not actually validated by experimental determination of inhibitory activity.

Based on these two facts I can not recommend for publication.

There initial steps, RF Model, descriptors selection are as well not so convincing, that they actually are completely novel or relevant (something above standard approach).

Authors should decide what they believe is important for scientific community, there computation approach, in the case they should focus on it and perhaps publish it is more specialized journal, or medicinal approach, where computational approach is only a tool, and focus on obtained results, in this case that requires experimental work in the lab.

Author Response

1. At the end authors use Autodoc Vina for selecting final compounds, which is very standard approach.

What the reviewer said was perfectly right. We have revised this issue (see lines 336-339).

It is a common method to select the final compound only by using AutoDock Vina software. However, the innovation of this paper lies not in the development and application of new molecular docking software, but in the following three points:

(1) Ligand-based drug design (e.g. machine learning) and structure-based drug design (e.g. molecular docking) are integrated for virtual screening and prediction of inhibitors. These two methods usually play their respective roles in drug discovery, but this research combined the two methods, which is seldom seen.

(2) In the field of drug design, machine learning methods are usually limited to the classification and regression of inhibitors, and rarely used in large-scale virtual screening. In this way, it can be used to screen unknown compound libraries and the built-in predictive probability score of the algorithm can be used for further screening. It is very effective to improve the hit rate of virtual screening, and the screening method in this paper can also be extended to virtual screening of large-scale databases.

(3) Many studies utilized free, open-source, on-line descriptors computing softwares or commercial softwares to calculate and predict the physical and chemical properties of inhibitors. In this paper, self-designed descriptors computing program and variable selection method are used to accurately describe the drug-like properties of molecules, such as the number of hydrogen bond donors, hydrogen bond acceptors and rotatable bonds. It can also filter a large number of descriptors to simplify the model, and it has an equally good prediction effect compared with other softwares, too.

According to the above innovations, although this paper used the very common AutoDock Vina software, it is innovative to use it in combination with machine learning methods, not to use this software alone to predict inhibitors. At the same time, this paper also aimed to observe and analyze the interaction between ligand and protein by AutoDock Vina software, and verify whether the virtually screened molecules through machine learning methods can produce relatively good binding effects with protein. After all, drug design based on machine learning only focuses on the structures of a large number of ligands, trying to find common ground to build general models, but ignoring the structure of receptor protein, let alone the interaction modes between these ligands and protein.

Multiple molecular docking methods such as Autodock Vina, ArgusLab, Molegro Virtual Docker, and Hex-Cuda have been used successfully by Mohammadi [1], to study the effect of alteration in the structure of carbamate-based acetylcholine esterase (AChE) inhibitors. The results of all tested methods showed good agreement. Uba [2] screened against class I HDACs from a large compound library containing 2,703,000 compounds by exhaustive approach of structure-based virtual screening using rDOCK and Autodock Vina to identify isoform-selective inhibitors, and a total of 41 compounds were found to show high-isoform selectivity. Ahmad et al. [3] found the best docked inhibitor compound-331 (6-(4-((3-methoxyphenylsulfonamido) methyl) phenyl)-2-methylnicotinamide)) in the aspects of GOLD fitness score and AutoDock Vina binding energy after comparative analysis. All of these literatures, which were not actually validated by experimental determination of inhibitory activity, combined AutoDock Vina with other softwares to identify inhibitors with specific targets, demonstrating the universality, high efficiency and reliability of the AutoDock Vina tool.

2. Selected compounds are then not actually validated by experimental determination of inhibitory activity.

The reviewer was absolutely right. The molecules screened in this paper have not been tested to verify their real biological activity. But on the one hand, the length of the article is limited, we can write another article on the whole series of work about lead optimization, chemical synthesis, structural characterization and biological testing of the selected molecules. It seems that the focus of the discussions is not prominent enough if we put the two researches together in this paper, and it will occupy too many pages, which does not meet the requirements of general papers. On the other hand, according to the so-called saying -- “every subject has its own experts”, our team's advantage is to study machine learning algorithms, code compilation and modification, and apply them to the field of drug design, describing molecular structures and physical and chemical characteristics, but for the fields of chemical synthesis, structural characterization and biological testing, we are just absolute blockheads. Of course, in the next research, our team will also consider cooperating with teams of organic synthesis and biological activity testing to work together for the discovery of new drug molecules. Our final result, six potentially new molecules for Top1 inhibition, will provide other scientists starting points for biochemical validation.

3. There initial steps, RF Model, descriptors selection are as well not so convincing, that they actually are completely novel or relevant (something above standard approach).

The reviewer was quite right. Feature selection methods are indeed very common in machine learning-based drug design, but they are very effective for analyzing the contribution of different characteristics to inhibitor activity. In this paper, a random forest program developed by our laboratory is used for feature selection, and the contribution of each descriptor to inhibitor activity can be scored, so that some features with higher ranking can be selected for comparison and analysis, it also can provide theoretical basis for the design of new drug molecules and the optimization of lead compounds. The random forest procedure and feature selection method adopted in this paper have published many articles and verified by several magazines [4-7], so the results are convincing. As for innovation, different feature selection methods have different algorithms and advantages. As long as the rank of features can be confirmed according to scores, the feature selection method should be considered as a new one. Moreover, the prediction accuracy of the random forest program in this paper reaches 97.12%, which is superior to the results in other articles [8]. It can prove that the feature selection method under this program has certain credibility in some way.

In recent studies, Qin et al. [9] applied machine learning methods, support vector machine and random forest, to develop 12 classification models based on 2925 diverse COX-2 inhibitors collected from 168 pieces of literature. The best MCC value of the external test set was predicted to be 0.68 by the RF model using ECFP_4 fingerprints. And they also identified substructures important for activity by descriptors selection process. Wang [10] used the C4.5, random forest (RF) and support vector machine (SVM) statistical methods for modeling and predicting thyroid hormone receptor (TR) agonists and antagonists, and the RF model possesses an average prediction accuracy of 84.0 and 87.1% for the cross-validation and external validation, respectively, demonstrating that RF and SVM models are useful tools capable of classifying TR-binding ligands as agonists or antagonists.

In the life sciences, the most benefited fields are cheminformatics, computational genomics and biomedical imaging [11]. Machine learning techniques, which belong to cheminformatics, ranging from the classification of substances (as active or inactive) to the construction of regression models and the ranking/virtual screening of databased compounds, could serve as useful tools for compound screening prior to synthesis and providing guidance for future experiment. For all these reasons, RF model, as one of the machine learning techniques, is a powerful approach in early drug discovery. The ability of RF to classify active or inactive compounds has enabled the prioritization of substances for virtual screening.

4. Authors should decide what they believe is important for scientific community, there computation approach, in the case they should focus on it and perhaps publish it is more specialized journal, or medicinal approach, where computational approach is only a tool, and focus on obtained results, in this case that requires experimental work in the lab.

Thai et al. [8] performed a support vector machine model on a series of 73 analogues to classify Benzo[c]phenanthridine derivatives according to TOP-I inhibitory activity. The best SVM model with total accuracy of 93% for training set was achieved using a set of 7 descriptors identified from a large set via a random forest algorithm. Sun et al. [12] performed a quantitative structure activity relationship (QSAR) and classification study based on a total of 134 base analogs related to their ED50 values (50% inhibitory concentration) against O6-methylguanine-DNA methyltransferase. Classification models were generated by seven machine-learning methods based on six types of molecular fingerprints. The two researches, which are very similar to our study, have been published in the journal “Molecules”. In addition, our work calculated 189 descriptors for each molecule. Molecular descriptors are numerical representations encoding aspects of the chemical information of a molecule, which can represent the molecule either by physicochemical properties, structural fragments or molecular fingerprints. Based on these two facts, I think our paper is suitable for this journal.

References

[1] Mohammadi T, Ghayeb Y. Atomic insight into designed carbamate-based derivatives as acetylcholine esterase (AChE) inhibitors: A computational study by multiple molecular docking and molecular dynamics simulation[J]. Journal of Biomolecular Structure and Dynamics, 2018, 36(1): 126-138.

[2] Uba A I, Yelekçi K. Identification of potential isoform-selective histone deacetylase inhibitors for cancer therapy: A combined approach of structure-based virtual screening, ADMET prediction and molecular dynamics simulation assay[J]. Journal of Biomolecular Structure and Dynamics, 2018, 36(12): 3231-3245.

[3] Ahmad S, Raza S, Abbasi S W, et al. Identification of natural inhibitors against Acinetobacter baumanniid-alanine-d-alanine ligase enzyme: A multi-spectrum in silico approach[J]. Journal of Molecular Liquids, 2018, 262: 460-475.

[4] Yang X G, Lv W, Chen Y Z, et al. In silico prediction and screening of γsecretase inhibitors by molecular descriptors and machine learning methods[J]. Journal of Computational Chemistry, 2010, 31(6): 1249-1258.

[5] Li B K, Cong Y, Tian Z Y, et al. Predicting and virtually screening the selective inhibitors of MMP-13 over MMP-1 by molecular descriptors and machine learning methods[J]. Acta Physico-Chimica Sinica, 2014, 30(1): 171-182.

[6] Li B K, He B, Tian Z Y, et al. Modeling, predicting and virtual screening of selective inhibitors of MMP-3 and MMP-9 over MMP-1 using random forest classification[J]. Chemometrics and Intelligent Laboratory Systems, 2015, 147: 30-40.

[7] Li B, Hu L, Xue Y, et al. Prediction of matrix metal proteinases-12 inhibitors by machine learning approaches[J]. Journal of Biomolecular Structure and Dynamics, 2019, 37: 2627-2640.

[8] Thai K M, Nguyen T Q, Ngo T D, et al. A support vector machine classification model for benzo [c] phenathridine analogues with topoisomerase-I inhibitory activity[J]. Molecules, 2012, 17(4): 4560-4582.

[9] Qin Z, Xi Y, Zhang S, et al. Classification of cyclooxygenase-2 inhibitors using support vector machine and random forest methods[J]. Journal of Chemical Information and Modeling, 2019, DOI: 10.1021/acs.jcim.8b00876.

[10] Wang F, Xing J. Classification of thyroid hormone receptor agonists and antagonists using statistical learning approaches[J]. Molecular Diversity, 2019, 23(1): 85-92.

[11] Maltarollo V G, Kronenberger T, Espinoza G Z, et al. Advances with support vector machines for novel drug discovery[J]. Expert Opinion on Drug Discovery, 2019, 14(1): 23-33.

[12] Sun G, Fan T, Sun X, et al. In silico prediction of O6-methylguanine-DNA methyltransferase inhibitory potency of base analogs with QSAR and machine learning methods[J]. Molecules, 2018, 23(11): 2892.

Reviewer 2 Report

Zhang et al describe an procedure based on different machine learning algorithms with the aim to identify potential Topoisomerase 1 inhibitors out of a screening compound database. Best ranked potential binders were further analysed by docking.

The idea of applying machine learning models for filtering of compound libraries is interesting and reliably methods for hit identification is a hot area in drug research. The manuscript is well written and most of the methods have been described properly. The presented final outcome, six potentially new molecules for Top1 inhibition, will provide other scientists starting points for biochemical validation.

I order to further improve the quality of the manuscript I recommend a minor revision based on the issues:

1a) Figure 1A: Please make the figure wider, as at the moment it is hard to see the facts that you describe in the text.

1b) Figure 1B: If the rainbow-like colorcoding does not yield any additional informaton, I would like to recommend to remove the rainbow effect.  

2) Figure 4: Quality have to be improved so that the descriptors can be read. If not possible enlarge the Figure and put it in the Supplementary. Please remove the rainbow color, as it does not provide any additional information.

3) Figure 5: Size can be reduced

4) Line 295: "The calculation of the descriptor was relied on the 3D structure... and the computing program was created by our lab."  Can you add some more details on that software?

5)Line 298: The authors state taht Corina software was used for 2D->3D. Please add version and reference for that software and add if standard parameters were used for conversion to allow reproduction of the results. 

6) Line 329: "For more than 50 years, Maybridge has been the forefront of innovative building blocks...". This is rather advertisment than science. Please remove it as there are lots of other vendors of screening compounds.

7) Line 349: "3D structure of each ligand was drawn by Chem,3D Pro 14.0." a)  I would recommend to exchange "drawn" with the term "calculated". b) I am confused as you state in Line 298 that Corina was used for 2D->3D conversion. Can you clarify that please?

8) The manuscript would benefit from an English native speaker style adjustment/correction.

Author Response

1a) Figure 1A: Please make the figure wider, as at the moment it is hard to see the facts that you describe in the text.

The Figure 1A had been enlarged, we’ve tried our best to make it clearer, which can be seen in line 154.

1b) Figure 1B: If the rainbow-like colorcoding does not yield any additional informaton, I would like to recommend to remove the rainbow effect.

Yes, the rainbow-like colorcoding does not yield any additional information. We have revised this issue (also see line 154).

2) Figure 4: Quality have to be improved so that the descriptors can be read. If not possible enlarge the Figure and put it in the Supplementary. Please remove the rainbow color, as it does not provide any additional information.

Before, we just wanted to emphasize the superiority of the RF model, so we list almost all the important descriptors of it. Now we put them into the Supporting Information (Table S2), only left 10 features in the Figure 4, and the rainbow color was also removed. It can be checked out in lines 197-202 and the Supporting Information.

3) Figure 5: Size can be reduced

Yes, it is too big for showing a little information. Thanks for the suggestion. It is in line 214 now.

4) Line 295: "The calculation of the descriptor was relied on the 3D structure... and the computing program was created by our lab."  Can you add some more details on that software?

The descriptors computing program was written in the Fortran 77 language, which is available for running under Linux environment. We have added this detail in lines 296 and 297.

5) Line 298: The authors state that Corina software was used for 2D->3D. Please add version and reference for that software and add if standard parameters were used for conversion to allow reproduction of the results.

We have revised this issue (see lines 299-302).

6) Line 329: "For more than 50 years, Maybridge has been the forefront of innovative building blocks...". This is rather advertisment than science. Please remove it as there are lots of other vendors of screening compounds.

We are not mean to advertise for Maybridge, we just want to show its advantage, and make the research result more credible. Now we have removed it to make it more scientific (see lines 326-330).

7) Line 349: "3D structure of each ligand was drawn by Chem,3D Pro 14.0."

a)  I would recommend to exchange "drawn" with the term "calculated".

Thanks for the advice. We truly neglected this issue. It can be seen in line 350 now.

b) I am confused as you state in Line 298 that Corina was used for 2D->3D conversion. Can you clarify that please?

We have revised this issue (see lines 299-302).

Reviewer 3 Report

-The x-axis and y-axis must only have one origin. There are two zeros along the x-axis. Kindly correct.

-The text on the x-axis of Figure 4 cannot bee read. Kindly enlarge. Do the same with Figure 9.

-Enlarge the labels on the amino acid residues of Figure 8.

Author Response

1. -The x-axis and y-axis must only have one origin. There are two zeros along the x-axis. Kindly correct.

The ROC curve figure was drawn by the IBM SPSS Statistics 19, and it was originally to be that way, it is more convenient to see the curve clearly. Of course we could make the origin points of x-axis and y-axis meet at 0. See for details in line 191.

2. -The text on the x-axis of Figure 4 cannot be read. Kindly enlarge. Do the same with Figure 9.

Yes, we wanted to show as more information as possible, but ignored the length of figures. Now they have been changed into bigger and clearer ones, which you can see in line 201 and line 274.

3. -Enlarge the labels on the amino acid residues of Figure 8.

We made their backgrounds to be white, and it is easy to see the amino acid residues in line 263.

Reviewer 4 Report

The authors present a combination of machine learning models with virtual screening and molecular docking to identify inhibitor compounds to the Topoisomerase I.

The paper is well written and clear, and the subject well presented. 

I would propose the authors to comment on the size of the training and testing sets and its influence in the results here presented. 

How different, qualitatively, are the results applying machine learning in combination with virtual screening and molecular docking, versus applying just virtual screening and docking regarding the detection of molecules with inhibition properties? Of course features selection would not be possible without ML.

I would suggest the authors improving the methods section of the virtual screening and molecular docking, as it would be hard to consider Autodock Vina, as a new docking program.

lines 341-342 - "Firstly, the downloaded file with a suffix “.pdb1” must be converted into a ".pdb" suffix by PyMol,..." This state is not true as the structure can be downloaded in pdb format directly from the website.

line 344: "...was selected to ensure ligand molecules were completely encapsulated.". Would this not be referring to the receptor? Otherwise, please clarify this statement.

lines 350-351: "They were processed with minimum energy,..." Please provide more information as this statement does not provide much information.

Author Response

1. I would propose the authors to comment on the size of the training and testing sets and its influence in the results here presented.

The size of the training and testing sets can be measured by molecular weight, which was shown in lines 271-276, ranging from 300 Da to 600 Da for 70% of compounds. An external validation set including 55 Top1is and the molecules picked out by virtual screening with RF scores greater than 0.7, are with molecular weights at reasonable levels, ranging from 200 Da to 500 Da, which means that the models developed according to the training set, the evaluation results confirmed by the testing set and the external validation set, and the potential Top1is predicted based on machine learning and virtual screening, are reasonable and reliable.

2. How different, qualitatively, are the results applying machine learning in combination with virtual screening and molecular docking, versus applying just virtual screening and docking regarding the detection of molecules with inhibition properties? Of course features selection would not be possible without ML.

Computer-aided drug design has become widely employed with several methods currently used to discover and develop new bioactive compounds, such as molecular docking, pharmacophore modeling, similarity analysis and quantitative structureactivity relationships, to mention the most common approaches [1]. These methods could generally be divided into two categories: ligand-based drug design and structure-based drug design strategies. Both of these methods offer their own advantages and disadvantages, and they are not exclusive; instead, one useful strategy is to employ multiple associated approaches in different steps of drug development to cover the weakness and compensate for the computational costs of each method. Specifically, machine learning techniques are often used in association with calculations of molecular descriptors, which can represent the molecule either by physicochemical properties, structural fragments or molecular fingerprints.

So, applying machine learning in combination with virtual screening and molecular docking is a more effective and accurate strategy in drug design.

Besides, when machine learning is introduced, the process of virtual screening becomes faster. For example, applying just virtual screening and docking, a thousand molecules may make it over the course of twenty hours, while applying machine learning, it only need a few seconds.

[1] Maltarollo V G, Kronenberger T, Espinoza G Z, et al. Advances with support vector machines for novel drug discovery[J]. Expert Opinion on Drug Discovery, 2019, 14(1): 23-33.

3. I would suggest the authors improving the methods section of the virtual screening and molecular docking, as it would be hard to consider Autodock Vina, as a new docking program.

What the reviewer said was perfectly right. We have revised this issue (see lines 336-339).

4. lines 341-342 - "Firstly, the downloaded file with a suffix “.pdb1” must be converted into a ".pdb" suffix by PyMol,..." This state is not true as the structure can be downloaded in pdb format directly from the website.

Thank you for offering this advice, so we tried that way, it worked, and we also changed that sentence in line 343.

5. line 344: "...was selected to ensure ligand molecules were completely encapsulated.". Would this not be referring to the receptor? Otherwise, please clarify this statement.

This statement referred to the original ligand CPT. We have revised this issue, which can be seen in line 344.

6. lines 350-351: "They were processed with minimum energy,..." Please provide more information as this statement does not provide much information.

We have revised this issue (see lines 350).

Round 2

Reviewer 1 Report

I carefully read the few changes in the manuscript and carefully read long reply to the reviewer, which did not describe many changes, but authors exhaustively explained their initial design of the study. Although I fully respect their response, I need to say that from my perspective, revised manuscript is can not be recommended for publication. My previous comment that still remains the same, authors should decide what they believe is important for scientific community, there computation approach, in the case they should focus on it and perhaps publish it is more specialized journal, or medicinal approach, where computational approach is only a tool, and focus on obtained results, in this case that requires experimental work in the lab.